# Drivers of associations between daytime-nighttime compound temperature extremes and mortality in China
Jun Yang [1,13] ✉, Maigeng Zhou[2,13], Cui Guo[3], Sui Zhu[4], Mohammad Javad Zare Sakhvidi[5], Weeberb J. Requia [6], Qinghua Sun[7], Shilu Tong[8,9,10], Mengmeng Li[11] & Qiyong Liu [12] ✉

## Abstract

**Background** Temperature extremes are anticipated to become more frequent and more intense under the context of climate change. While current evidence on health effects of compound extreme temperature event is scarce.

**Methods** This nationwide cross-sectional study collected daily data on weather and mortality for 161 Chinese districts/counties during 2007-2013. A quasi-Poisson generalized linear model was first applied to assess effects of daytime-only, nighttime-only and compound daytime-nighttime heat wave (and cold spell) on cause-specific mortality. Then a random-effect meta-analysis was used to produce pooled estimates at national level. Stratification analyses were performed by relative humidity, individual and regional characteristics.

**Results** Here we show that mortality risks of compound daytime-nighttime temperature extremes are much higher than those occurring only in the daytime or nighttime. Humid weather further exaggerates the mortality risk during heat waves, while dry air enhances the risk during cold weather. People who are elderly, illiterate, and those with ischemic heart disease and respiratory disease are particularly vulnerable to extreme temperature. At the community-level, population size, urbanization rate, proportion of elderly and PM2.5 are positively associated with increased risks associated with heat waves. Temperature, humidity and normalized difference vegetation index are positively associated with the effects of cold weather, with an opposite trend for latitude and diurnal temperature range.

**Conclusions** This nationwide study highlights the importance of incorporating compound daytime-nighttime extreme temperature events and humid conditions into early warning systems and urban design/planning.

## Plain language summary

Ongoing climate change has exaggerated the frequency and intensity of severe climate events, leading to substantial health and socioeconomic consequences. We assessed deaths in China during periods when many extreme climate events occurred at the same or similar times. We looked at deaths occurring during periods when both daytime and nighttime temperatures were very hot or cold. We found more serious health effects were seen when temperatures remained hot or cold during the day and night compared to when it was just hot or cold during the day or night. Other factors including humidity, preexisting heart or respiratory disease and age also impacted the risk of death. Our study highlights the detrimental health effects of many extreme climate events occurring together and the need for both people and governments to consider approaches to reduce these negative effects.

Under the context of climate change, extreme climate events are anticipated to become more frequent and sereve[1], which lead to excess morbidity and mortality[2-4]. For instance, over 70,000 excess deaths in European countries were attributed to the 2003 European heat wave[5]; three episodes of heat wave that occurred from June to August, 2017 in China led to 16,299 cumulative all-cause excess deaths[6]. In addition, the 2008 cold spell in 15 subtropical provinces of China caused 148,279 excess deaths[7,8]. An exceptional cold spell in February 2012 induced 1,578 additional deaths among those aged 75 years or older in 14 Italian cities[9]. Therefore, it is particularly vital to understand adverse health risks of extreme weather events in order to implement actionable response plan to protect human health.

Despite increasing evidence of detrimental health risks from temperature extremes (cold or heat)[2,3,10,11], distinct impacts of different types of temperature extremes, have received relatively little attention. Previous studies focusing solely on the daytime (or nighttime) temperature extreme might overestimate/underestimate the impact of these events by overlooking the confounding influence of the following nighttime (or daytime) exposure. Recent studies attempted to differentiate health impacts of temperature extremes between the daytime and nighttime period[12,13]. However, failing to consider the sustained period of extreme temperature events, these studies could not provide direct evidence on the difference in the health consequences of daytime, nighttime, and daytime-nighttime heat wave and

cold spell. Additionally, the influence of humidity on the associations between compound temperature extreme and health is also limited.

To gain insight into the knowledge gap above, we assess the mortality risks of different types of temperature extremes (heat wave and cold spell) in 161 Chinese communities by cause of death, personal characteristics, urban-rural region and temperature zone. We also explore the effect of relative humidity and city-level characteristics. We find that people who are elderly, illiterate, and those with ischemic heart disease and respiratory disease are particularly vulnerable to extreme temperature. Population size, urbanization rate, proportion of elderly and $PM_{2.5}$ are positively associated with increased risks associated with heat waves at the community level.

## Methods

### Study area and data sources

In this nationwide time-series research, daily mortality data (daily count) were collected in 161 Chinese communities (97 counties and 65 districts) between 2007 and 2013 from the Disease Surveillance Points system (DSPs) in China, covering 73 million people. A detailed description of this database and quality control procedure can be found elsewhere[14]. The study sites were classified into six climate zones: tropical, subtropical, Qinghai-Tibet alpine, warm temperature, mid temperature and cold temperature zone (Supplementary Fig. 1).

The ICD-10 (International Classification of Diseases, 10th version) was applied to classify the causes of death, containing non-accidental causes (ICD-10: A00–R99), cardiovascular disease (I00–I99), ischemic heart disease (IHD, I20–I25), stroke (I60–I69), respiratory disease (J00–J99) and chronic obstructive pulmonary disease (COPD, J40–J47). Daily number of deaths due to non-accidental cause were further divided by gender, age group (0–74 years and 75years or older) and educational level (people who are illiterate, and primary school or above).

Data on the daily meteorological variables between 2007 and 2013, including maximum, minimum and mean temperatures, air pressure, and relative humidity were derived from the China Meteorological Data Service Center (http://data.cma.cn/). We first interpolated the daily meteorological variable from 839 weather stations into 1 km×1 km resolution by the inverse distance weighting interpolation method to dilute the potential bias from a single station. Then, daily weather factor for each community was computed by aggregating the gridded values in its boundary.

Daily levels of fine particulate matter ($PM_{2.5}$) at 36 km×36 km resolution were firstly predicted using the modified community multiscale air quality model. Predicted $PM_{2.5}$ presented a strong correlation with monitoring data from 422 sites across 60 Chinese large cities[15,16]. The concentration of $PM_{2.5}$ for each community was computed by aggregating the value of grid cells in its boundary[17].

In order to explore city-level effect modifiers, we further collected community-level demographic data in 2010 from the China's 2010 population census, along with gross domestic product (GDP) per capita from the city-level or county-level statistical yearbook. The demographic data include total population size, percentage of urban residents, percentage of college or above, percentage of migrant residents, gender ratio (the ratio of male and female). Normalized difference vegetation index (NDVI) data on 16-days at 1-km resolution were downloaded from the Moderate Resolution Imaging Spectroradiometer NASA's Earth Observing System (https://modis.gsfc.nasa.gov/data/dataprod/mod13.php). After removing invalid values defined as lower than -0.2, the average NDVI value in 2010 for each community was calculated.

### Exposure definition

18 heat wave definition were firstly developed by combining nine relative thresholds (75.0th, 77.5th, 80th, 82.5th, 85th, 87.5th, 90th, 92.5th and 95.0th) and two durations of ≥2 and ≥3 days, and 18 cold spell definitions by combining nine relative thresholds (25.0th, 22.5th, 20th, 17.5th, 15.0th, 12.5th, 10th, 7.5th and 5.0th) and two durations of ≥2 and ≥3 days. Then, three types of extreme temperature events were defined: (1) an independent daytime-only event: only daytime but not the preceding nighttime with heat

wave or cold spell; (2) an independent nighttime-only event: only nighttime but not the following daytime with a heat wave or cold spell; (3) a sustained event: both daytime and the preceding nighttime with a heat wave or cold spell. To identify which definitions of heat wave and cold spell could capture their health impact better, the Akaike Information Criterion for quasi-Poisson (Q-AIC) was utilized to evaluate the goodness of model fits among different definitions[11]. The average value of Q-AIC from group-specific mortality was calculated. The minimum average of Q-AIC represented the best definition for heat wave and cold spell in each province. Days with extreme temperature events in conjunction with low (or high level) of humid condition was further classified with the median value of relative humidity at national level as cutoff.

### Statistical analysis

The mortality risks of heat wave or cold spell at daytime only, nighttime only and daytime-nighttime were estimated through a two-stage analytic approach. The data analyses of heat wave were restricted to the hot season (May to September) and cold spell to the cold season (November to March).

In the first stage, we used the quasi-Poisson generalized linear model allowing for over-dispersion to assess the community-specific estimate of extreme temperature event on mortality, with adjustment of the following confounders. A natural cubic spline function with 4 degrees of freedom (df) for day of the season was applied to capture the seasonality of daily death, and an indicator variable for the year was utilized to adjust for the long-term trend. Categorical variables for day of the week and public holidays were also included in the model to adjust for the potential variations in the week and holidays. Lagged effects of meteorological variables were captured by the two-dimensional cross-basis function produced distributed lag non-linear model, containing a natural cubic spline with 5 df for meteorological variable and a natural cubic spline with 4 df for the lag (up to 10 days). Model specifications for the covariates are consistent with many previous investigations[2,10,11,18]. The dummy variables of heat wave and cold spell were then separately introduced into the model. To reveal the lag patterns in the mortality risks of extreme temperature events, we fitted the models using different single lags (lag 0 day to lag 21 day), and moving average method to capture the cumulative lag effects.

In the second stage, a meta-analysis based on the residual maximum likelihood estimation was used to produce the pooled effects of extreme temperature events on mortality across 161 Chinese communities. The heterogeneity across the study sites was tested by $I^2$ statistic and Cochran's $Q$ method[19].

### Effect modification

Furthermore, we also explored potential effect modifications of relative humidity, personal-level and community-level characteristics on the associations of extreme temperature events with mortality. Stratification analyses with the two-stage analytic approach were conducted by the level of humidity, cause, gender, age group, educational attainment and region. The difference in the effects between subgroups was tested using the Z-test[2,20]. Finally, meta-regression analysis was further utilized to identify the community-level modifiers, and quantify the extent of heterogeneity explained by these variables.

### Sensitivity analysis

In order to test the robustness of the modeling parameters, several sensitivity analyses were performed by altering the df (from 3 to 5) and the maximum lag of days (from 8 to 21) for air pressure and humidity, and the df (from 4 to 8) for day of the year In addition, we also provided the effect estimates based on the commonly used definitions for heat wave (at least two consecutive days with daily temperature measures ≥ 90th percentile) and cold spell (at least two consecutive days with daily temperature measures ≤ 10th percentile). Finally, $PM_{2.5}$ was adjusted by a natural cubic spline with 3 df using data between 2011 and 2013, because $PM_{2.5}$ data were only available within this time period. Data manipulation were conducted using R

**Table 1 | Descriptive statistics of daily amounts of deaths and weather conditions between 2007 and 2013 in 161 Chinese communities**

| Variables | Mean | SD | Minimum | P25 | P50 | P75 | Maximum |
|---|---|---|---|---|---|---|---|
| Cause | | | | | | | |
| Non-accidental mortality | 7.3 | 6.1 | 0 | 3 | 6 | 10 | 142 |
| Cardiovascular mortality | 3.4 | 3.4 | 0 | 1 | 2 | 5 | 73 |
| IHD mortality | 1.2 | 1.6 | 0 | 0 | 1 | 2 | 44 |
| Stroke mortality | 1.8 | 2.0 | 0 | 0 | 1 | 3 | 48 |
| Respiratory mortality | 1.1 | 1.5 | 0 | 0 | 1 | 2 | 32 |
| COPD mortality | 0.8 | 1.3 | 0 | 0 | 0 | 1 | 28 |
| Gender | | | | | | | |
| Male | 4.2 | 3.8 | 0 | 2 | 3 | 6 | 86 |
| Female | 3.1 | 2.9 | 0 | 1 | 2 | 4 | 61 |
| Age (years) | | | | | | | |
| 0–74 | 3.8 | 3.4 | 0 | 1 | 3 | 5 | 86 |
| 75+ | 3.5 | 3.5 | 0 | 1 | 2 | 5 | 76 |
| Educational level | | | | | | | |
| Illiterate | 2.9 | 3.3 | 0 | 1 | 2 | 4 | 52 |
| Primary school or higher | 4.1 | 3.9 | 0 | 1 | 3 | 6 | 96 |
| Weather | | | | | | | |
| Maximum temperature (ºC) | 18.9 | 11.3 | −28.1 | 11.1 | 21.1 | 27.9 | 41.3 |
| Mean temperature (ºC) | 13.4 | 11.3 | −32.3 | 5.8 | 15.3 | 22.5 | 35.3 |
| Minimum temperature (ºC) | 9.1 | 11.7 | −36.3 | 1.5 | 10.7 | 18.4 | 31.4 |
| Relative humidity (%) | 65.8 | 18.5 | 0 | 54 | 69 | 80 | 100 |
| Air pressure (hPa) | 9508.5 | 900.5 | 6101 | 9167 | 9920 | 10073 | 32766 |

*SD* standard deviation, *IHD* ischemic heart disease, *COPD* chronic obstructive pulmonary disease.

software (version 3.5.1) with the "dlnm"[21] and "metafor"[22] packages. A two-sided $P < 0.05$ was viewed as statistical significance.

### Ethics
The present study involved only secondary analysis of daily aggregated and deidentified data, and is classified as exempt from IRB approval in the Chinese legal documents on ethics review that was issued by the National Health Commission of the People's Republic of China (document no. 4; 2023)[23].

### Reporting summary
Further information on research design is available in the Nature Portfolio Reporting Summary linked to this article.

## Results
### Study population characteristics
Table 1 presents descriptive information on daily amounts of deaths and daily weather variables between the year of 2007 and 2013 in 161 Chinese communities. In total, 2,742,717 non-accidental deaths, 1,274,558 cardiovascular deaths and 397,738 respiratory deaths were recorded in this study, with a corresponding daily average number of 7.4, 3.4 and 1.1 at the community level, respectively. The annual average value of daily minimum temperature and daily maximum temperature were 9.1 ℃ (ranging from −36.3 ℃ to 31.4 ℃) and 18.9 ℃ (from −28.1 ℃ to 41.3 ℃) at national level, respectively. During the study period, the annual average numbers at daytime only, nighttime only and daytime-nighttime was 3.9, 4.2 and 2.3 for heat wave, respectively; and 3.5, 3.6 and 3.1 for cold spell, respectively (Supplementary Table 1).

### Selection of heat wave and cold spell definition
Supplementary Tables 2 and 3 presents the Q-AIC value among 18 heat wave and 18 cold spell definitions. And Supplementary Data 1 provide the best heat wave and cold spell definitions for each community, with the lowest Q-AIC value. Thus, we mainly reported the results based on these identified heat wave and cold spell definitions.

### Lag structures of temperature extremes
Supplementary Fig. 2 presents the lag patterns of temperature extremes on non-accidental mortality. The impact of heat wave was generally significant at the current day and the second day, while the effect of cold spell was delayed during lag 0–3 days but persisted up to two weeks. Therefore, in the following findings, we mainly reported the cumulative effects of heat wave across lag 0–1 days but cold spell across lag 0–14 days.

### Effect modification by relative humidity
Heat wave at daytime, nighttime and daytime-nighttime was associated with an increase of 2.82% (95%CI: 1.21, 4.46), 1.16% (95%CI: −0.75, 3.10) and 8.86% (95%CI: 6.82, 10.94) in non-accidental mortality, respectively; the corresponding effect estimate of cold spell was 7.79% (95%CI: 1.71, 14.23), 11.94% (95%CI: 6.31, 17.86) and 16.25% (95%CI: 13.11, 19.47) in non-accidental mortality, respectively. When in conjunction with humid condition, effect estimates of heat wave were relatively higher at high levels of relative humidity, but higher effect estimates of cold spell were observed at low level of relative humidity, although most of the difference tests were not significant (Table 2 and Supplementary Table 4).

### Effect modification by cause, individual and climate
Figure 1 shows the mortality risks of heat wave across lag 0–1 days by cause, individual characteristics, region and climate. Effect estimates among the subgroups reveal a consistent tendency, with the highest impact from daytime-nighttime heat wave, followed by heat wave at daytime only and nighttime only. The highest effect estimates among causes of death were observed for IHD mortality during daytime-nighttime heat wave at 13.44%

**Table 2 | Percentage change (%) in mortality risk of temperature extremes at daytime only, nighttime only and daytime-nighttime, stratified by level of relative humidity**

| Temperature extreme | Subcategory | Total | Relative humidity | |
|---|---|---|---|---|
| | | | Low level | High level |
| Heat wave | Daytime only | 2.82 (1.21, 4.46) | 1.29 (−2.48, 5.20) | 3.23 (1.69, 4.78) |
| | Nighttime only | 1.16 (−0.75, 3.10) | 0.20 (−1.81, 2.24) | 1.31 (−1.06, 3.74) |
| | Compound | 8.86 (6.82, 10.94) | 7.59 (5.68, 9.54) | 10.66 (7.58, 13.84) |
| Cold spell | Daytime only | 7.79 (1.71, 14.23) | 15.40 (4.69, 27.2) | 6.44 (−0.27, 13.61) |
| | Nighttime only | 11.94 (6.31, 17.86) | 14.78 (8.69, 21.22) | 12.00 (1.61, 23.46) |
| | Compound | 16.25 (13.11, 19.47) | 22.07 (17.65, 26.65) | 17.69 (12.64, 22.97) |

The effects of heat wave and cold spell were estimated at lag 0–1 days and lag 0–14 days, respectively.

(95%CI: 8.50, 18.60). For the individual characteristics, the elderly, and people who were illiterate were more susceptible to the heat wave, particularly for the compound daytime-nighttime heat wave (all the difference tests between subgroups were statistically significant) (Supplementary Tables 5 and 6). For the temperature zone, relatively higher heat wave effects were observed among the people living in warm and mid-temperature zones.

Figure 2 presents the mortality risks of cold spell across lag 0–14 days by cause, individual characteristics, region and climate. The highest effect estimate among the causes of death was found for respiratory mortality during a daytime-nighttime cold spell at 24.18% (95%CI: 18.53, 30.09). And the vulnerable populations of cold spell were consistent with those from heat wave, except for region and temperature zone (ie, more substantial effect among the people in subtropical zone, and no difference between urban and rural) (Supplementary Tables 7 and 8). The results of heat wave and cold spell at lag 0 day were provided in Supplementary Tables 9 and 10.

### Sources of heterogeneity
Considering significant spatial heterogeneity among 161 Chinese communities (heat wave: Cochran's $Q = 274.62$, $I^2 = 40.37\%$, $P < 0.001$; cold spell: Cochran's $Q = 424.78$, $I^2 = 62.84\%$, $P < 0.001$), we further explored the community-level modifiers on the mortality risk of daytime-nighttime heat wave and cold spell. Among these factors, latitude, total population size, urbanization rate, proportion of the elderly and $PM_{2.5}$ concentrations were significantly and positively in association with increased mortality risk of heat wave. And annual mean temperature, humidity and normalized difference vegetation index (NDVI) were positively associated with risk of cold spell, but opposite trends for latitude and diurnal temperature range (Tables 3 and 4).

### Sensitivity analysis
The mortality risks of heat wave and cold spell were little influenced by altering the $df$ (3 to 5) for relative humidity and atmospheric pressure, the maximum lag from 8 to 21 days, and the $df$ (4 to 8) for the time trend (Supplementary Tables 11 and 12). When additionally adjusted for $PM_{2.5}$ using the data from 2011 to 2013, the estimates of daytime-nighttime heat wave at lag 0–1 day and cold spell at lag 0–14 day were 8.67% (95%CI: 6.58, 10.80) and 16.22 (95%CI: 13.14, 19.38), respectively, indicating the robustness of the major results. And slightly higher effect estimates were observed for the heat wave that was defined as at least two consecutive days with daily temperature measures (minimum temperature or maximum temperature) ≥ 90th percentile and cold spell as at least two consecutive days with daily temperature measures ≤ 10th percentile (Supplementary Tables 13 and 14).

### Discussion
In this nationwide modeling study, we found that daytime-nighttime temperature extremes were more harmful than those daytime- or nighttime-only events. Humid weather seemed further exaggerate the mortality risks of heatwave, while the risks of cold spell were likely enhanced by dry air. The older and persons with lower educational attainment were more susceptible to heat wave and cold spell, particularly during the prolonged period of extreme temperature. We also revealed that the hazardous effect of heat wave was stronger among people who lived in warm temperature zone, but more substantial cold spell effects on the people in the subtropical zone.

Heat wave and cold spell have been documented to be related to increased risks of morbidity and mortality[2,4,10,11,18,24], particularly for cardiorespiratory outcomes. The underlying mechanism linking harmful health effects of extreme weather events is less elucidated. For the cardiovascular system, exposure to sustained high temperature could cause a series of acute cardiovascular events when body thermoregulation is compromised or even failed. But when exposure to extreme low temperature, the supply of myocardial oxygen would be increased through activating the sympathetic nervous system and raising catecholamine, which might eventually cause myocardial infraction or ischemic heart attack[25,26]. For the respiratory outcomes, exposure to extreme high temperature could cause acute lung damage by producing pro-inflammatory cytokines (such as IL-1b and TNF-α). Moreover, cold temperature could cause airway epithelium damage, change the function and structure of the airway, and eventually aggravate the health conditions of patients with cardiorespiratory diseases.

We observed that the mortality risks of sustained temperature extremes were much higher than those daytime- or nighttime-only events (Supplementary Tables 15 and 16). By focusing only on the heat event defined as temperature over the 90th percentile but not excessive heat wave event, a previous study conducted in 98 Chinese communities also confirmed that the risks of sustained heat extremes on non-accidental mortality were markedly larger than that of daytime- and nighttime-only events[12]. The biological explanation of the health risk of sustained temperature extremes is unclear. During the sustained period of temperature extremes, people endure continuous exposure to extreme heat or cold, during which people's sleep time and sleep quality may be negatively impacted at night[27]. Then, the following daytime temperature extreme events would aggravate health conditions and cause lethal cardiovascular and respiratory diseases. In addition, our findings further indicated that the risks of heat wave were likely exaggerated by humid condition, while the risks of cold spell were likely enhanced by dry condition, which were consistent with previous studies[28,29]. Therefore, it is crucial to incorporate different types of temperature extremes into early warning systems and urban design/planning, and to strengthen the awareness of these adverse events from a public health perspective, as the frequency, intensity and duration of temperature extremes may dramatically increase within the context of climate change[30].

Our study found that the individual and community-related characteristics could modify the association between extreme weather events and mortality. These modification effects were more prominent during extreme temperature events occurred both daytime and nighttime. Previous studies have consistently reported higher vulnerability of heat wave and cold spell among the elderly[2,4,11]. Declined body functions in thermoregulation and homeostasis due to ageing, in conjunction with the high prevalence of existing chronic diseases, limited mobility and medication use, may lead to

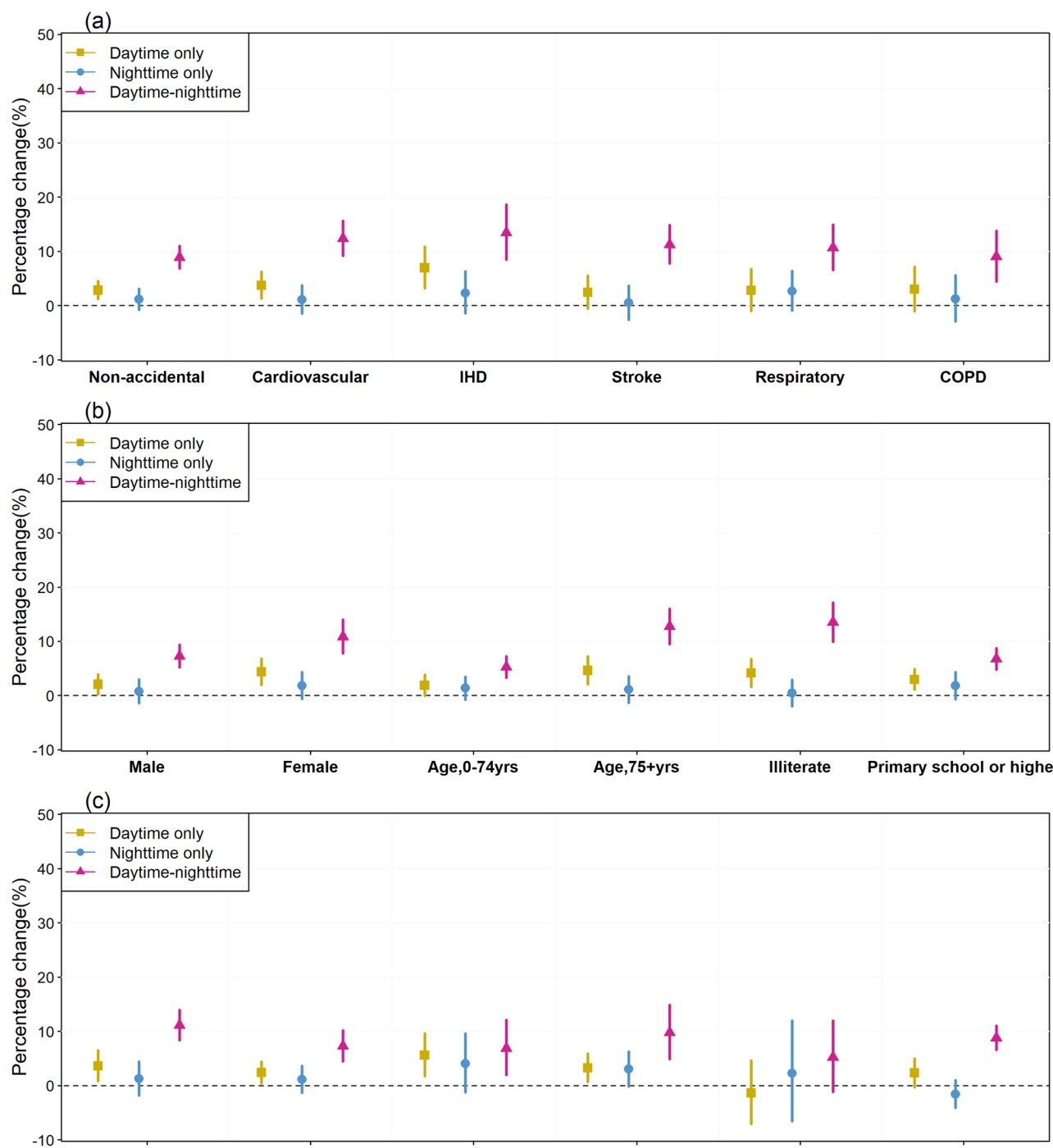

**Fig. 1 | Percentage change in mortality risk during multiple types of heat wave across lag 0–1 days.** The effect estimates during heat wave at daytime only, night-time only and daytime-nighttime across lag 0–1 days are stratified by cause (**a**), individual characteristics (**b**), and regions and climate (**c**). The square (or dot, triangle) and error bar denote the percentage and corresponding 95% confidence intervals.

the susceptibility of the seniors when exposed to temperature extremes. For educational level, people who were illiterate were confirmed to be at greater mortality risks from temperature extremes[2,31,32], which may be associated with poor working and living environments, and limited healthcare service and availability.

Furthermore, we revealed that people living in warm temperature zone were at higher risk from heat wave, while people in subtropical zone exhibited higher risk from cold spell. This tendency was further confirmed in the results of community-related effect modifiers that latitude was positively associated with heat wave effect but negatively associated with cold spell effect. And positive association between NDVI and risk of cold spell may be due to that the southern areas in China have higher NDVI than those norther areas, particularly during the cold season[33]. In addition, population residing in the community with lower diurnal temperature range were found to suffer from stronger effects of heat wave and cold spell. These phenomena can be partly explained by the human's physiological acclimatization to local or surrounding environment by long-term adaptation (such as dietary intake and building design), personal behaviors and technological equipment used. People residing in southern and warm regions are well prepared for the frequent high temperature, but not for cold

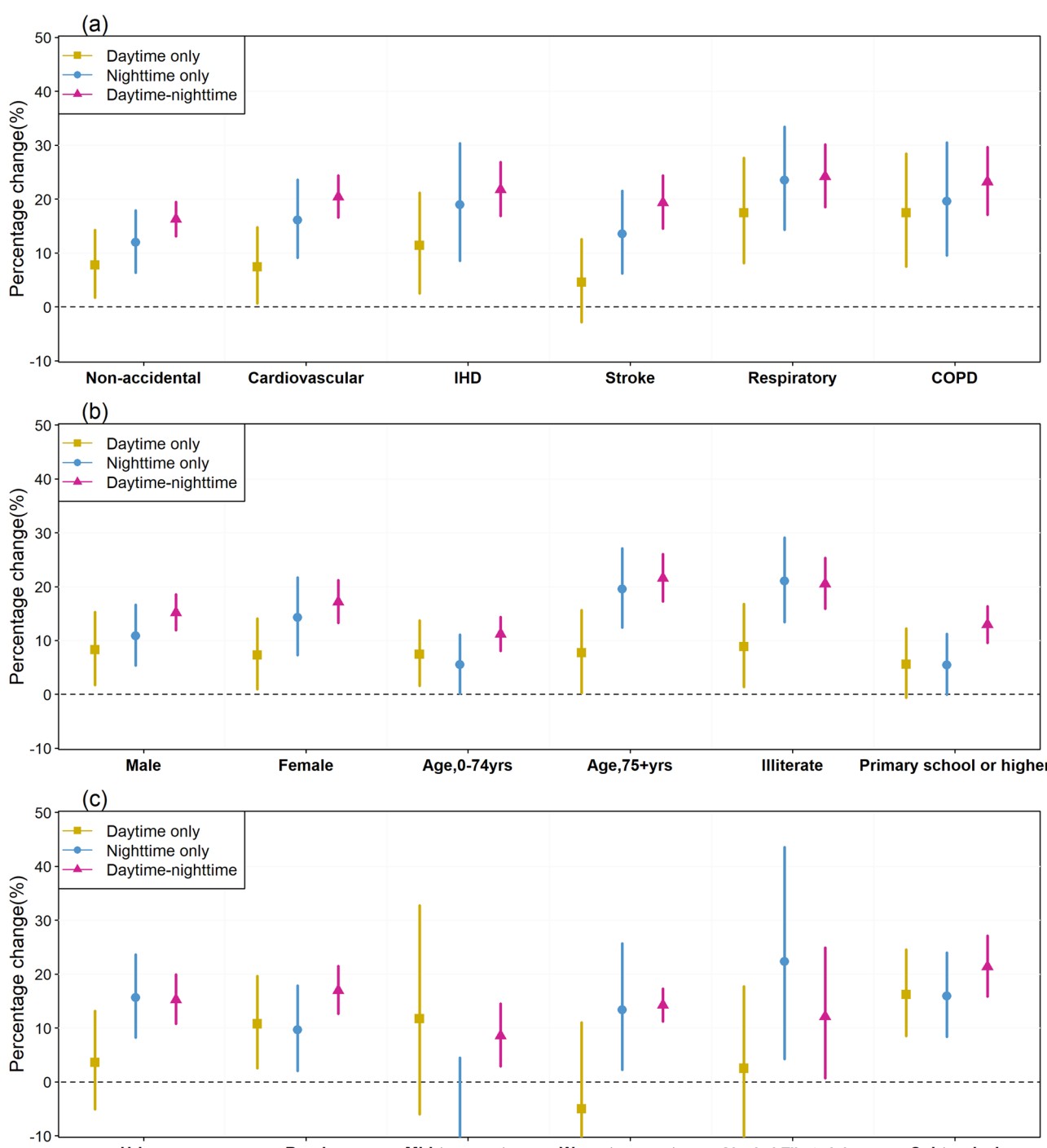

**Fig. 2 | Percentage change in mortality risk during multiple types of cold spell across lag 0–14 days.** The effect estimates during cold spell at daytime only, nighttime only and daytime-nighttime are stratified by cause (**a**), individual characteristics (**b**), and regions and climate (**c**). The square (or dot, triangle) and error bar denote the percentage and corresponding 95% confidence intervals.

temperatures and thus, they are more vulnerable to cold spell, while people in cold regions are less prepared for the high temperature and particularly sensitive to heat wave[3,34,35]. Similarly, people living in regions with low diurnal temperature range were not well prepared for days with the large change in temperature (always occurred during heat wave and cold spell)[20]. Therefore, education on the adverse health influences of temperature extremes and relevant proactive actions to these events is warranted in order to increase local people's resilience to high-risk climate events. Protective individual behaviors, including reducing outdoor physical activity, adequate intaking fluid and wearing light cloths during heat wave, and wearing layers

of fleece, wool or sythetic fabric and thick socks during cold spell, are a few examples to be recommended for the public.

Interestingly, we observed that residents in districts or counties with higher PM$_{2.5}$ concentrations had a greater vulnerability to heat wave but not cold spell, which is in line with previous studies[11,36]. Particulate matter and high temperature act on similar pathophysiological pathways, such as growing markers of systemic inflammation and oxidative stress. Therefore, it is biologically reasonable that particulate matter and high temperature may produce synergistic health effects[36]. Additionally, enhanced thermo-regulation during the days with high temperature such as increase in

**Table 3 | Percentage change (%) in the association between heat wave during 0–1 days and non-accidental mortality for per IQR increase in community-level predictors**

| Variable | IQR | Estimate | P value | Heterogeneity parameter ($\tau^2$) | Percentage of total variance due to between-study variance (%) |
|---|---|---|---|---|---|
| Latitude (°) | 9.8 | 0.24 (−2.89, 3.47) | 0.883 | 0.0059 | 45.69 |
| Population | 419484 | 2.01(0.15, 3.92) | 0.035 | 0.0055 | 44.39 |
| GDP per capita (RMB) | 81640 | −0.40 (−2.00, 1.23) | 0.626 | 0.0058 | 45.61 |
| Urbanization rate (%) | 42.2 | 3.57 (0.59, 6.63) | 0.019 | 0.0053 | 43.35 |
| Rate of elderly (%) | 2.7 | 4.02 (1.35, 6.76) | 0.003 | 0.0050 | 42.23 |
| College rate (%) | 11.2 | 1.98 (−0.18, 4.17) | 0.072 | 0.0056 | 44.57 |
| Literacy rate (%) | 4.2 | 0.01 (−2.12, 2.20) | 0.991 | 0.0057 | 45.40 |
| Migrant rate (%) | 23.3 | 2.55 (−0.40, 5.59) | 0.091 | 0.0056 | 44.61 |
| Temperature (°C) | 7.8 | 1.32 (−2.08, 4.84) | 0.451 | 0.0058 | 45.58 |
| Diurnal temperature range(°C) | 3.3 | −4.57 (−7.53, −1.51) | 0.004 | 0.0051 | 42.34 |
| Humidity (%) | 13.9 | 1.52 (−1.16, 4.27) | 0.270 | 0.0058 | 45.41 |
| $PM_{2.5}$ (µg/m³) | 44.6 | 3.58 (0.24, 7.03) | 0.036 | 0.0054 | 43.87 |
| NDVI | 0.3 | 0.24 (−2.89, 3.47) | 0.883 | 0.0059 | 45.69 |

IQR inter-quartile range, GDP gross domestic product, NDVI normalized difference vegetation index.

**Table 4 | Percentage change (%) in the association between cold spell during 0–14 days and non-accidental mortality for per IQR increase in community-level predictors**

| Variable | IQR | Estimate | P value | Heterogeneity parameter ($\tau^2$) | Percentage of total variance due to between-study variance (%) |
|---|---|---|---|---|---|
| Latitude (°) | 9.8 | −8.85 (−12.31, −5.24) | 0.001 | 0.0124 | 55.78 |
| Population | 419484 | 0.54 (−2.11, 3.25) | 0.693 | 0.0168 | 63.14 |
| GDP per capita (RMB) | 81640 | −0.69 (−3.12, 1.79) | 0.580 | 0.0168 | 63.31 |
| Urbanization rate (%) | 42.2 | 1.74 (−2.62, 6.30) | 0.441 | 0.0176 | 64.17 |
| Rate of elderly (%) | 2.7 | 1.49 (−2.32, 5.44) | 0.449 | 0.0177 | 64.39 |
| College rate (%) | 11.2 | −0.33 (−3.46, 2.9) | 0.838 | 0.0179 | 64.43 |
| Literacy rate (%) | 4.2 | −1.99 (−5.29, 1.42) | 0.248 | 0.0177 | 64.43 |
| Migrant rate (%) | 23.3 | 0.45 (−3.82, 4.92) | 0.839 | 0.0178 | 64.42 |
| Temperature (°C) | 7.8 | 10.52 (6.13, 15.09) | 0.001 | 0.0124 | 55.98 |
| Diurnal temperature range(°C) | 3.3 | −6.07 (−10.08, −1.87) | 0.005 | 0.0153 | 61.05 |
| Humidity (%) | 13.9 | 5.64 (1.67, 9.76) | 0.005 | 0.0153 | 60.91 |
| $PM_{2.5}$ (µg/m³) | 44.6 | −0.75 (−5.36, 4.08) | 0.755 | 0.0166 | 62.95 |
| NDVI | 0.3 | 5.19 (0.50, 10.11) | 0.030 | 0.0160 | 62.10 |

GDP gross domestic product, NDVI normalized difference vegetation index.

ventilation rate could consequently increase air pollution intake into the airways[37].

Moreover, we found that the community-level socioeconomic status could modify the health risk of temperature extreme. For instance, people in communities with higher population, urbanization rate and proportion of the elderly were at higher risk of heat wave. As discussed above that older people were more vulnerable to temperature extremes, it is reasonable that higher proportion of the elderly would increase the local's vulnerability of heat wave. Urbanization accompanied by the changes in land surface properties, including heat storage, soil moisture and albedo, contributes to higher temperature in urban core than the surrounding areas[38], also increase the trend of heat wave, and finally exaggerate the effect of heat wave on human health, particularly for those areas with higher population[39].

The present study has several strengths and public health implications. To the best of our knowledge, this is the largest scale investigation to estimate the mortality risk of compound extreme temperature events in China.

Moreover, the analysis covered 161 Chinese communities that were collected from a good representative database on disease surveillance and mortality in China. We observed more substantial effects of temperature extreme events that occurred both daytime and nighttime periods than those solely daytime or nighttime events. Our study highlights an urgent need to enhance the public awareness of and response to the hazards of compound extreme temperature events. The early warning system is suggested along with the tailored program and proper responses on adverse compound events to alert the local people about the preceding danger temperature events, particularly for the vulnerable populations.

Several limitations should be acknowledged in the present study. Firstly, similar with previous time-series studies[10,11], the data on weather conditions from fixed monitoring stations were instead of the individual exposure data, which may cause measurement errors in the exposure. Secondly, the classification of health outcomes was according to the ICD codes. There is a possibility of misclassification error. But this error seems to

be random and non-differential. Furthermore, during extreme temperature events, people are more inclined to stay indoors, which may lead to underestimating the true effects of extreme temperature events. Lastly, it is still challenge in determination of appropriate thresholds for heat wave and cold wave, especially for the countries with widely different climates (like China). In our preliminary analysis, we have attempted to identify the threshold by incorporating the model residuals from autoregressive integrated moving average (ARIMA) of daily non-accidental deaths with daily temperature, which are motivated by previous studies[40,41]. However, the mortality residuals obtained from ARIMA fluctuate of most locations fluctuate around the zero or are below the zero (Supplementary Method and Supplementary Figs. 3–6), preventing us from selecting the thresholds subjectively. This heterogeneity may be due to the widely different climates, various healthcare service ability and adaptation capacity in the sample sites of the present study. And future research is warranted to compare the predicting performance of thresholds determined by different approach and also to explore the potential influencing factors.

In conclusion, our study quantitatively revealed that day-night sustained temperature extremes were more harmful to human health than those solely daytime or nighttime events. Females, the elderly and people with lower educational attainment were more susceptible to both heat wave and cold spell. Besides, stronger impact of heat wave was observed among people living in warm temperature zone, while stronger effect of cold spell was found on people in subtropical zone. It is of great importance to incorporate different types of temperature extremes into early warning system and urban design/planning, and to strengthen the awareness of the health consequences of these events for the public health domain as climate change proceeds.

## Data availability

The dataset generated and analyzed during this study are available from the corresponding authors upon reasonable request. The mortality data can be obtained from the Chinese Center for Disease Control and Prevention (China CDC) by following the process explained at http://www.phsciencedata.cn/Share/en/index.jsp. Access is granted if users agree not to engage in unauthorized distribution of the raw data to a third party and to use the data for scientific research only. Historical temperature data are available at the dataset of daily climate data from Chinese surface stations of the China Meteorological Data Service Center by following the process explained at http://data.cma.cn/. Source data for the Figs. 1 and 2 are available as Supplementary Data 2 and Supplementary Data 3, respectively.

## Code availability

The code is available on GitHub[42].

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

## Acknowledgements
The study was supported by the National Natural Science Foundation of China (No. 82003552), Guangdong Basic and Applied Basic Research Foundation (No. 2020A1515011161) and the National Key Research and Development Program of China (No. 2018YFC0213600).

## Author contributions
J.Y. and Q.L. conceived the study. J.Y., M.Z., Q.L., M.L. and S.Z. collected the data. J.Y. and M.L. performed the data curation and formal analysis. J.Y. wrote the first draught of the paper. C.G., M.L., M.J.Z.S., W.J.R., Q.S., S.T. and Q.L. contributed to the critical revision of the manuscript. All authors read and approved the final manuscript.

## Competing interests
The authors declare no competing interests.

## Additional information

[1]School of Public Health, Guangzhou Medical University, Guangzhou 511436, China. [2]National Center for Chronic and Noncommunicable Disease Control and Prevention, Beijing 100050, China. [3]Department of Urban Planning and Design, Faculty of Architecture, The University of Hong Kong, Hong Kong SAR, China. [4]Department of Public Health and Preventive Medicine, School of Medicine, Jinan University, Guangzhou 510080, China. [5]Department of Occupational Health, School of Public Health, Shahid Sadoughi University of Medical Sciences, Yazd, Iran. [6]School of Public Policy and Government, Fundação Getúlio Vargas, Brasília, Distrito Federal, Brazil. [7]School of Public Health, Zhejiang Chinese Medical University, Hangzhou 310053, China. [8]Shanghai Children's Medical Center, Shanghai Jiao Tong University, Shanghai 200127, China. [9]School of Public Health and Institute of Environment and Population Health, Anhui Medical University, Hefei, China. [10]School of Public Health and Institute of Health and Biomedical Innovation, Queensland University of Technology, Brisbane, QLD, Australia. [11]State Key Laboratory of Oncology in South China, Guangdong Provincial Clinical Research Center for Cancer, Sun Yat-sen University Cancer Center, Guangzhou, China. [12]National Key Laboratory of Intelligent Tracking and Forecasting for Infectious Diseases, National Institute for Communicable Disease Control and Prevention, Chinese Center for Disease Control and Prevention, Beijing, China. [13]These authors contributed equally: Jun Yang, Maigeng Zhou. ✉e-mail: yangjun_eci@jnu.edu.cn; liuqiyong@icdc.cn

