## [Peer review File · Communications Medicine]

Drivers of associations between daytime-nighttime compound temperature extremes and mortality in ChinaReviewers' comments:

Reviewer #1 (Remarks to the Author):

The authors have performed a solid analysis to compare the association of daytime and/or nighttime temperature extremes in Chinese communities. It is very thorough, using well-established methods.

I have the following comments:

- The authors dwell on climate change a lot at the beginning of the abstract and the first paragraph of the introduction. But this is not a climate change paper, but rather a paper on environmental epidemiology where the focus is climate-relevant. Please tone down and reduce the references to climate change, as it is largely irrelevant and background context, not central focus.

- In the introduction, are there are previous works that can be quoted about China? The examples quoted are in countries with potentially quite different contexts (health system infrastructure etc.)

- Line 87: What is the spatial unit for these quoted numbers? By community?

- Lines 88-89: I would check these number ranges again as they seem incorrect.

- Can the authors provide a little more justification about why they use the 10th and 90th percentiles in the Methods? Currently they simply cite 2 and 11 without any comment, but this is not helpful to the average reader.

- Code should always be available on publication. I would recommend changing the code availability statement to state that the code will be published on a GitHub repo and somewhere with a DOI like Zenodo.

In summary, the authors have made a robust analysis on an interesting research question with very well established methods, so there is little to discuss methodologically.

Reviewer #2 (Remarks to the Author):

This is a manuscript that analyses the Drivers of associations between daytime-nighttime compound and mortality.

Although it is an interesting article, from my point of view it has an important conceptual problem that invalidates the results obtained in it.

The authors define the definitions of heatwave and coldwave based on fixed 10th percentile for coldwaves and 90th percentile for heatwaves.

The impact of heatwaves on mortality is influenced by many more factors than low temperatures (socio-economic, demographic, infrastructure, medical services, etc.), as the WHO points out in the case of heat. "The defining heat wave temperatures should be determined from epidemiological studies for each location and not from fixed meteorological percentiles", and the authors do not do this. It may be that the heatwave temperature in some places corresponds to the 84th percentile, but in another place it is the 95th percentile. In other words, setting a fixed percentile may be comparing heatwave mortality with mortality in other places that is not. The same applies to cold spells.

Therefore, the study as it stands is not fully valid and the results obtained are not fully valid either.

The authors should determine the definition temperature of heat wave and cold wave for each place as indicated by the WHO and then calculate the mortality burden attributable to these extreme temperatures.

I believe that publishing a manuscript on the health effects of heat waves and cold spells against the WHO guidelines lacks scientific validity.

Reviewers' comments:

Reviewer #1 (Remarks to the Author):

The authors have performed a solid analysis to compare the association of daytime and/or nighttime temperature extremes in Chinese communities. It is very thorough, using well-established methods.

Response: We greatly appreciate the reviewer's constructive critiques and comments. We have made point-to-point responses to these comments as follow, and highlighted the amendments in the revised manuscript.

I have the following comments:

- The authors dwell on climate change a lot at the beginning of the abstract and the first paragraph of the introduction. But this is not a climate change paper, but rather a paper on environmental epidemiology where the focus is climate-relevant. Please tone down and reduce the references to climate change, as it is largely irrelevant and background context, not central focus.

Response: We thank reviewer for this helpful comment. In order to focus on the topic of our study, we have replaced the sentence relating to “climate change” by “temperature extreme”-related contents at the beginning of the abstract and the first paragraph of the introduction as suggested.

- In the introduction, are there are previous works that can be quoted about China? The examples quoted are in countries with potentially quite different contexts (health system infrastructure etc.)

Response: In the revised manuscript, we have quoted the unprecedented 2008 cold spell and 2017 heat wave in China, which have respectively attributed to 148,279 and 16,299 excess deaths in China. Please refer to Lines 63-66 & Page 2.

- Line 87: What is the spatial unit for these quoted numbers? By community?

Response: Yes, it is at the community level. We have provided this information in Lines 92-93 & Page 3. Thank you.

- Lines 88-89: I would check these number ranges again as they seem incorrect.

Response: Sorry for this typo. We have carefully checked this issue throughout the manuscript.

- Can the authors provide a little more justification about why they use the 10th and 90th percentiles in the Methods? Currently they simply cite 2 and 11 without any comment, but this is not helpful to the average reader.

Response: Thank you for this comment. Considering your concern and other comments raised by another reviewer ("The defining heat wave temperatures should be determined from epidemiological studies for each location and not from fixed meteorological percentiles"), we first developed 18 heat wave definitions and 18 cold spell definitions for each community, and then identified the most appropriate one based on the goodness of model fits. The detailed process is listed as follow.

Firstly, we developed 18 heat wave definitions combining nine relative thresholds (75.0th, 77.5th, 80th, 82.5th, 85.0th, 87.5th, 90th, 92.5th and 95.0th) and two durations of ≥ 2 and ≥ 3 days, and 18 cold wave definitions by combining nine relative thresholds (25.0th, 22.5th, 20th, 17.5th, 15.0th, 12.5th, 10th, 7.5th and 5.0th) and two durations of ≥ 2 and ≥ 3 days for each community. Then, the Akaike Information Criterion for quasi-Poisson (Q-AIC) was utilized to evaluate the goodness of model fits among different definitions. The average value of Q-AIC from group-specific mortality was calculated. The minimum average of Q-AIC produced the best definition for heat wave and cold wave in each province (an average of 5 communities for each province). We select the best definitions of heat wave and cold wave at the provincial level because the communities at same

province generally have similar climate. And more importantly, the public health policy and early warning system for extreme temperature events that are developed at the provincial level are conducive to be implemented and managed.

Supplemental Table 2-4 present the identified definition of heat wave and cold wave for each community, with the lowest Q-AIC value among 18 heat wave and 18 cold wave definitions. And based on these definitions, we have re-conducted the analyses, and also updated the result and discussion section.

- Code should always be available on publication. I would recommend changing the code availability statement to state that the code will be published on a GitHub repo and somewhere with a DOI like Zenodo.

Response: The R-code will be published on GitHub before publication. Thank you.

In summary, the authors have made a robust analysis on an interesting research question with very well established methods, so there is little to discuss methodologically.

Response: Thank you for your constructive comments.

Reviewer #2 (Remarks to the Author):

This is a manuscript that analyses the Drivers of associations between daytime-nighttime compound and mortality.

Although it is an interesting article, from my point of view it has an important conceptual problem that invalidates the results obtained in it.

The authors define the definitions of heatwave and coldwave based on fixed 10th percentile for coldwaves and 90th percentile for heatwaves.

The impact of heatwaves on mortality is influenced by many more factors than low temperatures (socio-economic, demographic, infrastructure, medical services, etc.), as the WHO points out in the case of heat. "The defining heat wave temperatures should be determined from epidemiological studies for each location and not from fixed meteorological percentiles", and the authors do not do this. It may be that the heatwave temperature in some places corresponds to the 84th percentile, but in another place it is the 95th percentile. In other words, setting a fixed percentile may be comparing heatwave mortality with mortality in other places that is not. The same applies to cold spells.

Therefore, the study as it stands is not fully valid and the results obtained are not fully valid either.

The authors should determine the definition temperature of heat wave and cold wave for each place as indicated by the WHO and then calculate the mortality burden attributable to these extreme temperatures.

I believe that publishing a manuscript on the health effects of heat waves and cold spells against the WHO guidelines lacks scientific validity.

Response: We appreciate these insightful/constructive comments on the definition of heat wave and cold wave. And we agree that "The defining heat wave temperatures should be determined from epidemiological studies for each location and not from fixed meteorological percentiles". In order to identify the appropriate definition of heat wave and cold wave for each community, 18 heat wave definitions that combined nine relative thresholds (75.0th, 77.5th, 80th,

82.5th, 85.0th, 87.5th, 90th, 92.5th and 95.0th) and two durations of ≥ 2 and ≥ 3 days, and 18 cold wave definitions by combining nine relative thresholds (25.0th, 22.5th, 20th, 17.5th, 15.0th, 12.5th, 10th, 7.5th and 5.0th) and two durations of ≥ 2 and ≥ 3 days were first developed for each community. Then, the Akaike Information Criterion for quasi-Poisson (Q-AIC) was utilized to evaluate the goodness of model fits among different definitions. The average value of Q-AIC from group-specific mortality was calculated. The minimum average of Q-AIC produced the best definition for heat wave and cold wave in each province (an average of 5 communities for each province). We select the best definitions of heat wave and cold wave at the provincial level because the communities at same province generally have similar climate. And more importantly, the public health policy and early warning system for extreme temperature events that are developed at the provincial level are conducive to be implemented and managed.

Supplemental Table 2 presents Q-AIC value of 18 heat wave and cold spell definitions, and Supplemental Table 3-4 provide the identified definition of heat wave and cold wave for each community, with the lowest Q-AIC value. And based on these definitions, we have re-conducted the analyses, and also updated the result and discussion section.

Reviewers' comments:

Reviewer #1 (Remarks to the Author):

I am satisfied with the revisions the authors have made. Thank you for your additional work.

Reviewer #2 (Remarks to the Author):

Although the authors say that they agree with my assessment and that of the WHO in the first revision of the article that: "The defining heat wave temperatures should be determined from epidemiological studies for each location and not from fixed meteorological percentiles", in the revised version they still do not determine what is the defining heat wave and cold wave temperature from an epidemiological point of view and continue to base it on fixed percentiles.

In other words, they have not solved the methodological problem raised.

My decision is to reject the article until the heat wave definition temperature is calculated as established by the WHO.

The results obtained with this methodology have no scientific value.

Reviewers' comments:

Reviewer #1 (Remarks to the Author):

I am satisfied with the revisions the authors have made. Thank you for your additional work.

Response: We greatly appreciate the reviewer's positive comments.

Reviewer #2 (Remarks to the Author):

Although the authors say that they agree with my assessment and that of the WHO in the first revision of the article that: "The defining heat wave temperatures should be determined from epidemiological studies for each location and not from fixed meteorological percentiles", in the revised version they still do not determine what is the defining heat wave and cold wave temperature from an epidemiological point of view and continue to base it on fixed percentiles.

In other words, they have not solved the methodological problem raised.

My decision is to reject the article until the heat wave definition temperature is calculated as established by the WHO.

The results obtained with this methodology have no scientific value.

Response: We thank the reviewer for your valuable time and these important comments. To our knowledge, the reviewer has two major concerns regarding the heat wave (cold wave) definitions: determining definition for each location, and definition temperature established by the WHO. Therefore, in order to facilitate the reviewer to review our responses, we separately reply to these two points as follows.

1) Determining definition for each location

Actually, we have developed 18 heat wave definitions and 18 cold wave definitions for each community, separately added the extreme temperature

events identified by these definitions each time to the time-series model, adjusting for potential covariates, and finally selected the most appropriate heat wave and cold wave definition based on the Akaike Information Criterion for quasi-Poisson (Q-AIC). Specifically, 18 heat wave definitions that combined nine relative thresholds (75.0th, 77.5th, 80th, 82.5th, 85.0th, 87.5th, 90th, 92.5th and 95.0th) and two durations of ≥ 2 and ≥ 3 days, and 18 cold wave definitions by combining nine relative thresholds (25.0th, 22.5th, 20th, 17.5th, 15.0th, 12.5th, 10th, 7.5th and 5.0th) and two durations of ≥ 2 and ≥ 3 days were first developed for each community. Then, under each definition, we identified heat wave and cold wave events, and included these events separately into first stage time-series model (the most common use method for estimating acute effect of exposure in epidemiological studies) in each community, after full considering covariates (such as time trend of daily death, days of the week, holiday and weather factors). And the Q-AIC was utilized to evaluate the goodness of model fits among different definitions. The average value of Q-AIC from group-specific mortality was calculated. The minimum average of Q-AIC produced the best definition for heat wave and cold wave in each province (an average of 5 communities for each province). This approach is in line with many previous studies^[1-5]. We select the best definitions of heat wave and cold wave at the provincial level because the communities at same province generally have similar climate. And more importantly, the public health policy and early warning system for extreme temperature events that are developed at the provincial level are conducive to be implemented and managed. Supplemental Table 2-4 present the identified definition of heat wave and cold wave for each community, with the lowest Q-AIC value among 18 heat wave and 18 cold wave definitions.

Therefore, we have already determined heat wave definition and cold wave definition for each location.

2) *Definition temperature established by the WHO.*

We agree with reviewer that it is important to use the heat wave definition temperature established by the WHO. However, after a careful searching (using “heat wave”, “definition” and “world health organization” as keywords) on Google and PubMed, we were not able to find a practical guideline on heat wave (or cold spell) proposed by WHO. Taking PubMed as example, we only found two relevant publications (Figure A), but none of them have applied any guideline on heat wave definition by WHO. Instead, the percentiles (ie, 90th) were used as the threshold to define the heat wave in their papers.

Figure A. The searching result in PubMed by using “heat wave”, “definition” and “world health organization” as keywords.

Therefore, from the technical point, how to define the heat wave (and cold spell) proposed by the WHO is not very clear for us. If possible, we humbly hope that the reviewer could provide us more specific and detailed information, such as the references about the practical guideline on heat wave (and cold spell) definition proposed by WHO. We are happy to update the analysis when this technical puzzle is clear.

References

- [1] Chen J, Yang J, Zhou M, et al. Cold spell and mortality in 31 chinese capital cities: Definitions, vulnerability and implications. *Environment international*, 2019, 128: 271-278
- [2] Chen K, Bi J, Chen J, et al. Influence of heat wave definitions to the added effect of heat waves on daily mortality in nanjing, china. *Science of the Total Environment*, 2015, 506: 18-25
- [3] Dai M, Chen S, Huang S, et al. Increased emergency cases for out-of-hospital cardiac arrest due to cold spells in shenzhen, china. *Environ Sci Pollut Res Int*, 2023, 30: 1774-1784
- [4] Tian Z, Li S, Zhang J, et al. The characteristic of heat wave effects on coronary heart disease mortality in beijing, china: A time series study. *PLoS One*, 2013, 8: e77321
- [5] Yang J, Yin P, Sun J, et al. Heatwave and mortality in 31 major chinese cities: Definition, vulnerability and implications. *Science of The Total Environment*, 2019, 649: 695-702

Reviewers' comments:

Reviewer #2 (Remarks to the Author):

I thank the authors for their willingness to calculate heatwave and coldwave defining temperatures based on epidemiological studies and not on fixed climatic percentiles as recommended by the WHO.

In the linked manuscript, in which people from WHO are listed as co-authors, the calculation methodology is specified as requested by the authors. <https://doi.org/10.1016/j.envint.2017.11.012>.

Reviewer #2 (Remarks to the Author):

I thank the authors for their willingness to calculate heatwave and coldwave defining temperatures based on epidemiological studies and not on fixed climatic percentiles as recommended by the WHO.

In the linked manuscript, in which people from WHO are listed as co-authors, the calculation methodology is specified as requested by the authors. <https://doi.org/10.1016/j.envint.2017.11.012>.

Response: Thank you very much for providing us the reference on determining heat wave and cold wave threshold. Following the methodology raised by this publication, we applied the univariate autoregressive integrated moving average (ARIMA) for the daily non-accidental deaths, and then separately incorporate the residuals obtained from ARIMA with daily maximum temperature and minimum temperature at 1°C. Different from our expectation [threshold should be the temperature above (for heat wave) or below (for cold wave) which mortality residuals were consistently and significantly increased], however, the mortality residuals at high temperature in nearly half of the locations fluctuate around the zero (highlighted by red rectangles in Figure A for daily maximum temperature, and in Figure B for daily minimum temperature), or are below the zero (blue rectangles). Regarding these results, it is of great challenge for us to have a convincing and objective way to determine the heat wave (or cold wave) threshold for each location, especially when both thresholds of daily maximum and minimum temperature are needed to be considered for defining compound extreme temperature events in our study.

Therefore, we hope that we could retain our previous method used for determining definition for each location. Specifically, we developed 18 heat wave definitions and 18 cold wave definitions for each community, separately added the extreme temperature events identified by these definitions each time to the time-series model, adjusting for potential covariates, and finally selected the appropriate heat

wave and cold wave definition based on the Akaike Information Criterion for quasi-Poisson (Q-AIC). This method has also been widely used in epidemiological investigations, including our multicounty study [Guo Y, ..., Tong S. Heat Wave and Mortality: A Multicountry, Multicommunity Study. *Environ Health Perspect.* 2017; 125(8): 087006] and our multicity study in China [Yang J, ..., Liu Q. Heatwave and mortality in 31 major Chinese cities: Definition, vulnerability and implications. *Sci Total Environ.* 2019; 649: 695-702], and also used by other studies^[1-4].

But to be prudent, we have cited the suggestive reference and added the following limitation to the discussion section: It is still challenge in determination of the appropriate thresholds for heat wave and cold wave, especially for the countries with widely different climates (like China). In our preliminary analysis, we have attempted to identify the threshold by incorporating the model residuals from autoregressive integrated moving average (ARIMA) of daily non-accidental deaths with daily temperature, which were motivated by previous studies^[5, 6]. However, the mortality residuals obtained from ARIMA of most locations fluctuate around the zero or are below the zero, preventing us from selecting the thresholds subjectively. This heterogeneity may be due to the widely different climates, various healthcare service ability and adaptation capacity in the sample sites of the present study. And future research is still warranted to compare the predicting performance of thresholds determined by different approaches and also to explore the potential influencing factors (Lines 242-251 & Page 7).

References

- [1] Chen J, Yang J, Zhou M, et al. Cold spell and mortality in 31 chinese capital cities: Definitions, vulnerability and implications. *Environment international*, 2019, 128: 271-278
- [2] Chen K, Bi J, Chen J, et al. Influence of heat wave definitions to the added effect of heat waves on daily mortality in nanjing, china. *Science of the Total Environment*, 2015, 506: 18-25
- [3] Dai M, Chen S, Huang S, et al. Increased emergency cases for out-of-hospital cardiac arrest due to cold spells in shenzhen, china. *Environ Sci Pollut Res Int*, 2023, 30: 1774-1784
- [4] Tian Z, Li S, Zhang J, et al. The characteristic of heat wave effects on coronary heart disease mortality in beijing, china: A time series study. *PLoS One*, 2013, 8: e77321
- [5] Martinez GS, Diaz J, Hooyberghs H, et al. Heat and health in antwerp under climate change: Projected impacts and implications for prevention. *Environment International*, 2018, 111: 135-143
- [6] Montero JC, Mirón IJ, Criado JJ, et al. Comparison between two methods of defining heat waves: A retrospective study in castile-la mancha (spain). *Science of The Total Environment*, 2010, 408: 1544-1550

(continue)

Figure A. Barplot for determining heat wave definition on the basis of daily maximum temperature in 161 Chinese communities.

(continue)

Figure B. Barplot for determining heat wave definition on the basis of daily minimum temperature in 161 Chinese communities.

REVIEWERS' COMMENTS:

Reviewer #2 (Remarks to the Author):

am aware of the work involved in using temperature-mortality diagrams to detect heatwave-defining temperatures and although the authors still do not do so, at least there is an explicit mention of what the appropriate methodology should be. I hope that as they say they will apply it in further studies.

From my point of view the article can be published.

Reviewer #2 (Remarks to the Author):

am aware of the work involved in using temperature-mortality diagrams to detect heatwave-defining temperatures and although the authors still do not do so, at least there is an explicit mention of what the appropriate methodology should be. I hope that as they say they will apply it in further studies.

From my point of view the article can be published.

Response: We are pleased that you are satisfied with the revision that we made in last round, and recommend our article to be accepted for publication.

(continue)

Figure A. Barplot for determining heat wave definition on the basis of daily maximum temperature in 161 Chinese communities.

(continue)

Figure B. Barplot for determining heat wave definition on the basis of daily minimum temperature in 161 Chinese communities.